# IndustryGPT: A Large Language Model for Industrial Domain-Specific Question Answering

## Abstract

Large Language Models are rapidly being adapted for high-stakes professional domains, yet the industrial sector, with its demand for deep expertise and precision, presents unique and formidable challenges. Standard fine-tuning approaches are often insufficient. In this work, we identify a critical paradox in domain-specific SFT: enriching training data with detailed explanations or Chains-of-Thought can counter-intuitively degrade a model's factual accuracy, revealing a fundamental conflict between learning to be correct and learning to be verbose. To resolve this, we propose a novel two-stage fine-tuning framework that first anchors the model's core knowledge using direct question-answer pairs, and only then cultivates its advanced reasoning and explanatory abilities. To rigorously evaluate our method and establish a much-needed standard for the field, we introduce the Industry-QA Benchmark, a comprehensive dataset of over 10,000 questions spanning numerous industrial disciplines, which we will release to the community. We supplement this with a curated industrial subset from SuperGPQA to ensure robust and generalizable evaluation. Our resulting model, IndustryGPT, demonstrates state-of-the-art performance, significantly outperforming strong proprietary and open-source models on our benchmarks. Crucially, it achieves this specialized expertise without any degradation of its general capabilities. This work presents not only a superior model for the industrial domain but also a principled training methodology that resolves a key challenge in developing specialized AI.

## 1 Introduction

The ascent of Large Language Models (LLMs) like GPT-4 (OpenAI, 2023), Gemini (Team et al., 2023), and powerful open-source alternatives (Touvron et al., 2023; Bai et al., 2023) has redefined the boundaries of artificial intelligence. Built upon the Transformer architecture (Vaswani et al., 2017), these models exhibit a remarkable fluency in understanding, generating, and reasoning with natural language. This success has ignited a wave of interest in deploying them in high-stakes, specialized domains, particularly the industrial sector, where precision, reliability, and deep expertise are not just valued, but essential (Gao et al., 2024).

However, the journey from a general-purpose LLM to a trusted industrial expert is fraught with challenges. The most apparent hurdle is the **knowledge gap**; models trained on general web text simply lack the specialized vocabulary and nuanced understanding of industrial processes (Li et al., 2024). Yet, our investigation reveals a deeper, more counter-intuitive challenge that lies at the heart of the domain adaptation process itself. When attempting to fine-tune these models on expert-level data, we uncovered a significant paradox: training on rich data containing detailed explanations ('Question, Answer+Explanation') can actively harm the model's ability to select the correct answer, performing worse than models trained on simple question-answer pairs. This suggests a debilitating conflict between two competing learning objectives: the discriminative goal of being factually accurate and the generative goal of being fluently explanatory.

To resolve this fine-tuning paradox, we developed a **two-stage SFT framework**, which forms the technical core of our work. As illustrated in Figure 2, this method decouples the competing tasks by first anchoring the model's factual accuracy before cultivating its ability to reason and explain. Yet, proving the efficacy of such a nuanced approach is impossible without a dedicated evaluation

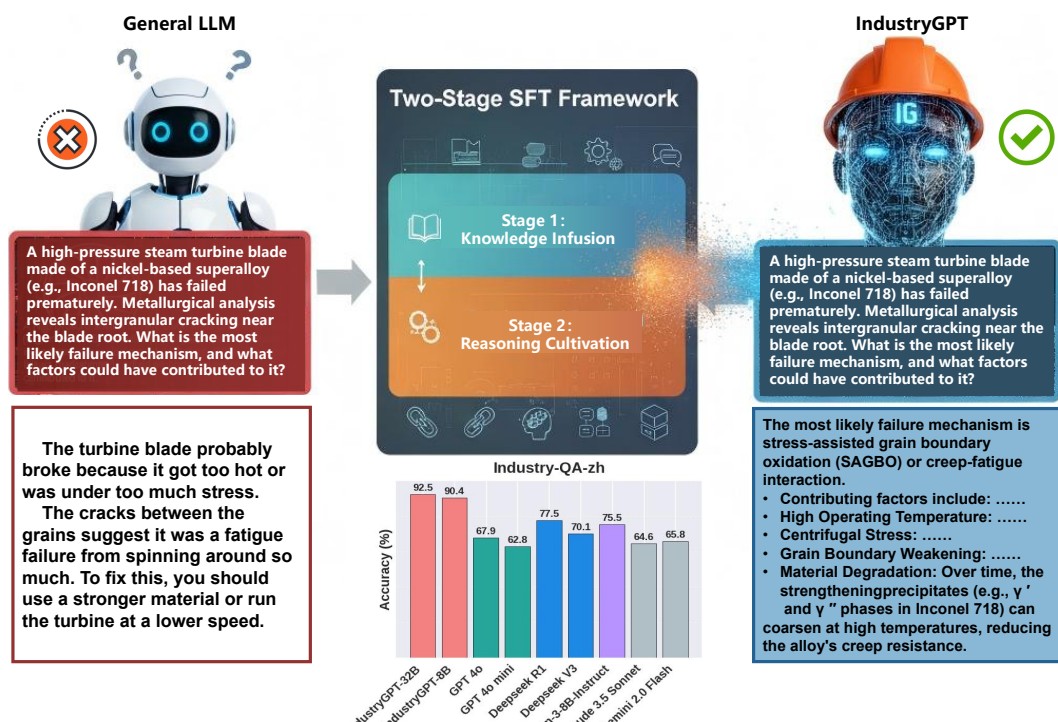

Figure 1: Advancing Industrial Intelligence with IndustryGPT. Our novel two-stage Supervised Fine-Tuning (SFT) framework transforms generic Large Language Models into IndustryGPT, a specialized powerhouse for industrial applications. The framework first infuses industrial knowledge (Stage 1) and then cultivates Reasoning capabilities (Stage 2) using Chain-of-Thought data. As exemplified, while a General LLM struggles with a complex turbine blade failure analysis, IndustryGPT provides precise, expert-level diagnoses and actionable insights. This specialized model achieves State-of-the-Art performance on our newly constructed Industry-QA Benchmark while meticulously preserving general abilities.

framework, which highlights another critical void: the **evaluation gap**. While fields like medicine (Jin et al., 2021) and law (Guha et al., 2023) have established benchmarks, the industrial domain lacks a comprehensive tool to rigorously measure the very capabilities we seek to build.

To holistically address these interconnected challenges of paradoxical training dynamics and inadequate evaluation, we introduce **IndustryGPT**. This paper details our systematic approach, with the following primary contributions:

- **A Two-Stage SFT Framework:** We propose a principled fine-tuning strategy that resolves the conflict between accuracy and explanation. The first stage, *Knowledge Infusion*, anchors the model's factual correctness, while the second, *Reasoning Cultivation*, builds upon this foundation to develop deep explanatory and problem-solving skills.
- **A Large-Scale Industrial Benchmark:** To enable robust evaluation, we construct and release the **Industry-QA Benchmark**, a comprehensive testbed of over 10,000 questions designed to assess both core knowledge and complex reasoning in diverse industrial disciplines.
- **State-of-the-Art Performance:** Through extensive experiments, we demonstrate that IndustryGPT significantly outperforms strong open-source and proprietary models, validating the superiority of our approach.
- **Preservation of General Capabilities:** We show that this deep domain specialization is achieved without catastrophic forgetting, as IndustryGPT maintains its core general reasoning abilities when evaluated on the challenging **MMLU-Pro** benchmark (Zhang et al., 2024).

## 2 RELATED WORK

### 2.1 FROM GENERALIST MODELS TO DOMAIN SPECIALISTS

The current era of AI is largely defined by powerful foundation models like GPT-4 (OpenAI, 2023), Llama (Touvron et al., 2023), and Qwen (Bai et al., 2023), which have demonstrated remarkable versatility. While these general-purpose models serve as potent starting points, unlocking their true potential in specialized fields requires targeted adaptation. The research community has explored several techniques to bridge this gap. Early efforts focused on **continued pre-training** on domain-specific corpora, successfully creating experts like BioBERT (Lee et al., 2020) for medicine and FinBERT (Araci, 2019) for finance by enriching their core vocabulary and conceptual understanding.

More commonly, models are adapted via **supervised fine-tuning (SFT)**, where they learn from curated sets of expert examples (Ouyang et al., 2022). To move beyond simple fact-retrieval and instill deeper problem-solving skills, many have turned to **Chain-of-Thought (CoT) fine-tuning**, training models to emulate the step-by-step reasoning processes of human experts (Wei et al., 2022; Ho et al., 2022). Our work builds directly on these principles but is motivated by a critical limitation we observed: naively mixing factual and reasoning-based data in a single SFT stage can lead to suboptimal performance. Our two-stage framework addresses this by systematically separating these concerns, first grounding the model in factual knowledge before cultivating its complex reasoning abilities.

### 2.2 THE NEED FOR SPECIALIZED EVALUATION

A specialized model is only as good as the benchmark used to validate it. As LLMs have been adapted for professional domains, a suite of robust evaluation tools has emerged. Fields like medicine now have rigorous tests such as MedQA (Jin et al., 2021) and PubMedQA (Jin et al., 2019), while the legal profession has LegalBench (Guha et al., 2023), and finance has benchmarks like FinQA (Chen et al., 2021). These tools are crucial for driving progress and ensuring reliability.

In stark contrast, the industrial domain has lacked a modern, comprehensive benchmark. While broad evaluations like MMLU (Hendrycks et al., 2021) contain engineering-related questions, they only scratch the surface of the practical, multi-step problems faced in manufacturing, diagnostics, and process control. This evaluation gap makes it difficult to meaningfully assess a model's readiness for real-world industrial deployment. The construction of our **Industry-QA Benchmark** is therefore a foundational contribution, designed specifically to fill this critical void and provide a dedicated testbed for the advanced capabilities required in industrial applications.

## 3 METHODOLOGY

This section details our approach to developing IndustryGPT. We begin by introducing the base models, then narrate the discovery process that led to our core contribution: a novel two-stage supervised fine-tuning framework. We conclude with a theoretical analysis that explains why this staged approach is superior for instilling both deep domain knowledge and sophisticated reasoning capabilities.

### 3.1 BASE MODELS

The foundation of IndustryGPT rests upon the Qwen series of large language models (Bai et al., 2023), renowned for their strong performance and architectural efficiency. We selected Qwen models for their competitive baseline capabilities, efficient Group Query Attention (GQA) architecture (Ainslie et al., 2023), and support for long context windows, a crucial feature for industrial applications. To investigate the effects of scale, we developed IndustryGPT variants based on two sizes: **Qwen3-8B** and **Qwen3-32B**. The 8B model offers an efficient balance of capability and accessibility, while the 32B model allows us to explore the upper limits of performance on complex industrial tasks.

## 3.2 A Two-Stage Framework for Knowledge and Reasoning

Our initial explorations into adapting these models involved standard supervised fine-tuning (SFT) on our collected industrial datasets. We experimented with two data formats: simple, direct 'Question, Answer' pairs, and richer 'Question, Answer+CoT' pairs that included a detailed reasoning chain. This led to the counter-intuitive finding shown in Table 1: while training on 'Answer+CoT' improved performance over the base model, it was significantly less effective than training on 'Answer Only'. The model, when forced to learn reasoning and answering simultaneously, seemingly compromised its factual accuracy.

Table 1: Initial SFT experiments on the Industry-QA benchmark (Accuracy %) using an 8B model. The results reveal that adding CoT explanations in a single stage harms answer accuracy.

| Model Configuration | Accuracy (%) |
|---|---|
| Base Model (Qwen3-8B-Instruct) | 80.2 |
| SFT on Answer with CoT | 84.9 |
| SFT on Answer Only | **89.0** |

Motivated by this observation, we hypothesized that the tasks of knowledge acquisition and reasoning cultivation are in conflict during a naive, single-stage SFT process. We therefore designed a **two-stage SFT framework** to decouple these competing objectives, as illustrated in Figure 2.

**Stage 1: Knowledge Infusion.** The first stage is designed to anchor the model's factual accuracy. We fine-tune the base model on a large corpus of over 200,000 industrial 'Q, A' pairs, mixed with a smaller portion of general-purpose conversation data (Taori et al., 2023) to maintain versatility. The goal is singular: create an intermediate model that has mastered the core knowledge of the industrial domain.

**Stage 2: Reasoning Cultivation.** Building on this knowledgeable foundation, the second stage focuses exclusively on teaching the model *how* to solve problems. We use a high-quality dataset of approximately 40,000 'Q, A+CoT' examples. Since the model from Stage 1 is already proficient at identifying the correct answer 'A', the optimization pressure in this stage naturally shifts to learning the conditional task of generating the reasoning 'E' given 'Q' and 'A'.

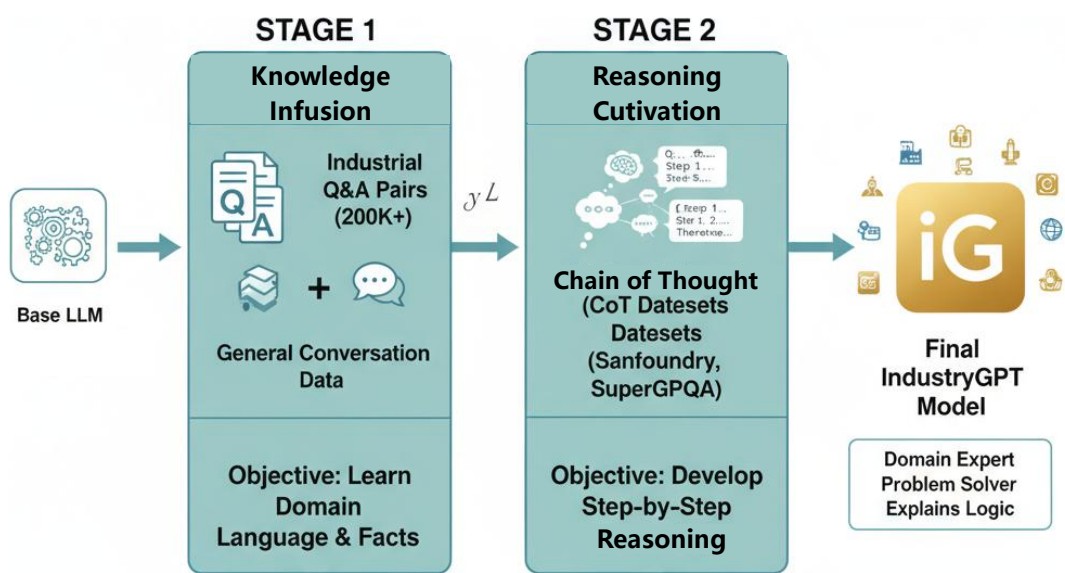

Figure 2: Overview of the Two-Stage SFT Framework. Motivated by the observation that simultaneously training on answers and CoT is suboptimal, our framework first anchors factual knowledge (Stage 1) before cultivating complex reasoning (Stage 2), leading to a more capable and accurate final model.

### 3.3 THEORETICAL JUSTIFICATION

The superiority of our two-stage approach, as empirically validated in our ablation studies, can be explained through the lens of optimization theory and curriculum learning. A naive, single-stage fine-tuning process on mixed data creates a complex optimization problem fraught with competing objectives, whereas our framework decouples these challenges into a more stable and effective learning sequence.

#### 3.3.1 THE CHALLENGE OF SINGLE-STAGE MIXED-DATA SFT

In a single-stage approach, the model is trained on data containing full explanations, 'Q, A+E'. For a given sample, the model's objective is to minimize the negative log-likelihood of the entire target sequence $(A, E)$. Using the chain rule of probability, this composite loss can be formally decomposed:

$$\mathcal{L}_{total}(Q, A, E; \theta) = -\log P(A, E|Q; \theta)$$
$$= \underbrace{-\log P(A|Q; \theta)}_{\mathcal{L}_{ans}} \underbrace{-\log P(E|Q, A; \theta)}_{\mathcal{L}_{exp}} \quad (1)$$

Here, the total loss $\mathcal{L}_{total}$ is effectively a sum of the loss for the answer, $\mathcal{L}_{ans}$, and the loss for the explanation, $\mathcal{L}_{exp}$. This formulation creates two primary challenges:

**1. Competing Objectives and Gradient Conflict.** The single objective masks two fundamentally different sub-tasks: a focused, *discriminative* task of minimizing $\mathcal{L}_{ans}$ by selecting the correct answer, and a broad, *generative* task of minimizing $\mathcal{L}_{exp}$ by composing a fluent explanation. The optimization landscapes for these tasks differ significantly. The gradients for each sub-task, $\nabla_\theta \mathcal{L}_{ans}$ and $\nabla_\theta \mathcal{L}_{exp}$, can point in conflicting directions, leading to destructive interference where progress on one task comes at the expense of the other. This conflict results in an unstable optimization path, making it difficult to converge to a solution that excels at both accuracy and explanation.

**2. Loss Scale Imbalance and Attention Dilution.** The explanation 'E' is typically much longer than the answer 'A'. Consequently, the magnitude of the loss signal from the numerous tokens in $\mathcal{L}_{exp}$ often dwarfs that of the few (often single) tokens in $\mathcal{L}_{ans}$. The optimization process becomes dominated by the goal of generating plausible-sounding text for the explanation, while the critical objective of ensuring the answer's factual correctness is diluted. This is exacerbated by the Transformer's attention mechanism; a longer target sequence can diffuse attention, potentially weakening the model's focus on the key information required to determine the correct answer.

#### 3.3.2 THE ADVANTAGE OF THE TWO-STAGE FRAMEWORK

Our two-stage framework mitigates these issues by reformulating the problem, aligning with the core principles of Curriculum Learning (Bengio et al., 2009), which posits that learning is more effective when concepts are presented in a simple-to-complex order.

**Stage 1: Anchoring Factual Accuracy.** The first stage isolates the "easy" task. The model optimizes a single, focused objective: finding the parameters $\theta_1$ that are highly specialized for answer accuracy.

$$\theta_1 = \arg\min_\theta \mathbb{E}_{(Q,A) \sim D_{ans}}[-\log P(A|Q; \theta)] \quad (2)$$

This focused training allows the model to converge to a favorable region in the parameter space where the core discriminative task is reliably solved. It effectively "anchors" the model's knowledge.

**Stage 2: Conditional Reasoning Cultivation.** The second stage begins with the well-initialized parameters $\theta_1$ and tackles the "harder" task. The objective is to find the final parameters $\theta_2$:

$$\theta_2 = \arg\min_\theta \mathbb{E}_{(Q,A,E) \sim D_{full}}[-\log P(A, E|Q; \theta)], \quad \text{where optimization starts from } \theta_1 \quad (3)$$

Because the model, initialized with $\theta_1$, already predicts $P(A|Q)$ with high confidence, the loss component $\mathcal{L}_{ans}$ and its corresponding gradient magnitude $||\nabla_\theta \mathcal{L}_{ans}||$ are small from the outset. The optimization pressure therefore naturally shifts to the conditional task of learning the explanation given the question and the (already known) correct answer, i.e., learning $P(E|Q, A)$. The gradient conflict is effectively neutralized, transforming the problem from learning a complex joint distribution into a simpler, sequential one. This ensures that both factual accuracy and explanatory reasoning are fully and robustly developed.

## 4 BENCHMARKS AND EXPERIMENTAL SETUP

This section introduces the evaluation framework for our study. We first detail the construction and composition of our primary contribution, the **Industry-QA Benchmark**, and then outline the full suite of benchmarks, baseline models, and the evaluation protocol used in our experiments.

### 4.1 THE INDUSTRY-QA BENCHMARK

To address the critical evaluation gap for LLMs in the industrial domain, we constructed the **Industry-QA Benchmark**. Motivated by the lack of practical, reasoning-focused tests in existing benchmarks like MMLU (Hendrycks et al., 2021), we sourced over 10,000 questions from authoritative materials, including professional engineering textbooks, industry certification exams, and technical manuals. This ensures that our benchmark reflects the real-world challenges faced by industrial professionals.

The final benchmark consists of **10,571** questions spanning 12 core industrial disciplines. As shown in Figure 3, it has a diverse subject distribution, with strong representation from core fields like Mechanical and Electrical Engineering. The questions are formatted as True/False, Single-Choice, and Multiple-Choice to facilitate robust, standardized evaluation. A detailed breakdown of the question format distribution is provided in Table 2.

Table 2: Key statistics and question format distribution of the Industry-QA Benchmark.

| Statistic | Value |
|---|---|
| Total Number of Questions | 10,571 |
| Number of Core Disciplines | 12 |
| *Question Format Distribution* | |
| True/False Questions | 4,323 (40.9%) |
| Single-Choice Questions | 4,357 (41.2%) |
| Multiple-Choice Questions | 1,891 (17.9%) |

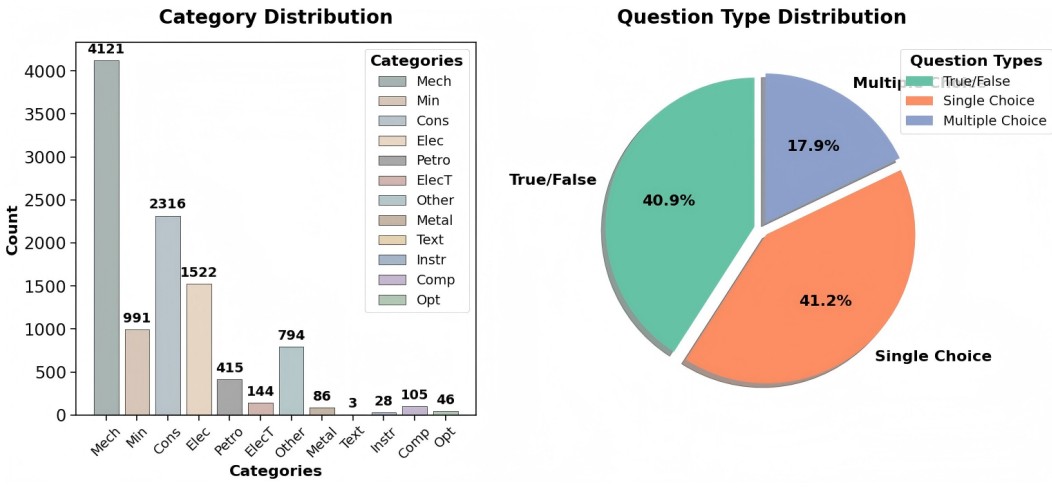

Figure 3: The composition of the Industry-QA benchmark. Left: Distribution of questions across 12 major industrial disciplines. Right: Breakdown of question formats.

### 4.2 EXPERIMENTAL PROTOCOL

**Evaluation Suite.** Our evaluation is designed to be comprehensive, assessing three key aspects:

- **Primary Domain Evaluation** on our bilingual **Industry-QA** benchmark and its challenging **Industry-QA-Hard** subset.
- **Generalization Assessment** on a curated industrial subset of the **SuperGPQA** benchmark (Liu et al., 2024).

- **General Capability Preservation** using the **MMLU** benchmark (Hendrycks et al., 2021).

**Baseline Models.** We compare IndustryGPT against a strong suite of models, including its base models (**Qwen3-Instruct**), leading open-source models (**Llama-3**, **DeepSeek**), and state-of-the-art proprietary models (**GPT-4o**, **Claude 3.5 Sonnet**).

**Evaluation Settings.** All models are evaluated in a zero-shot setting using a standardized instruction prompt. The primary metric is **Accuracy**. The fine-tuning process was conducted on NVIDIA A100 80GB GPUs using the PyTorch framework with DeepSpeed and FlashAttention-2 optimizations.

# 5 RESULTS AND ANALYSIS

We now present the results of our comprehensive experiments. The overall performance landscape is summarized in Figure 4, which provides a high-level comparison of IndustryGPT against baselines across all key benchmarks. We analyze these findings in detail below, covering domain-specific performance, the validation of our methodology through ablation studies, and the preservation of general capabilities.

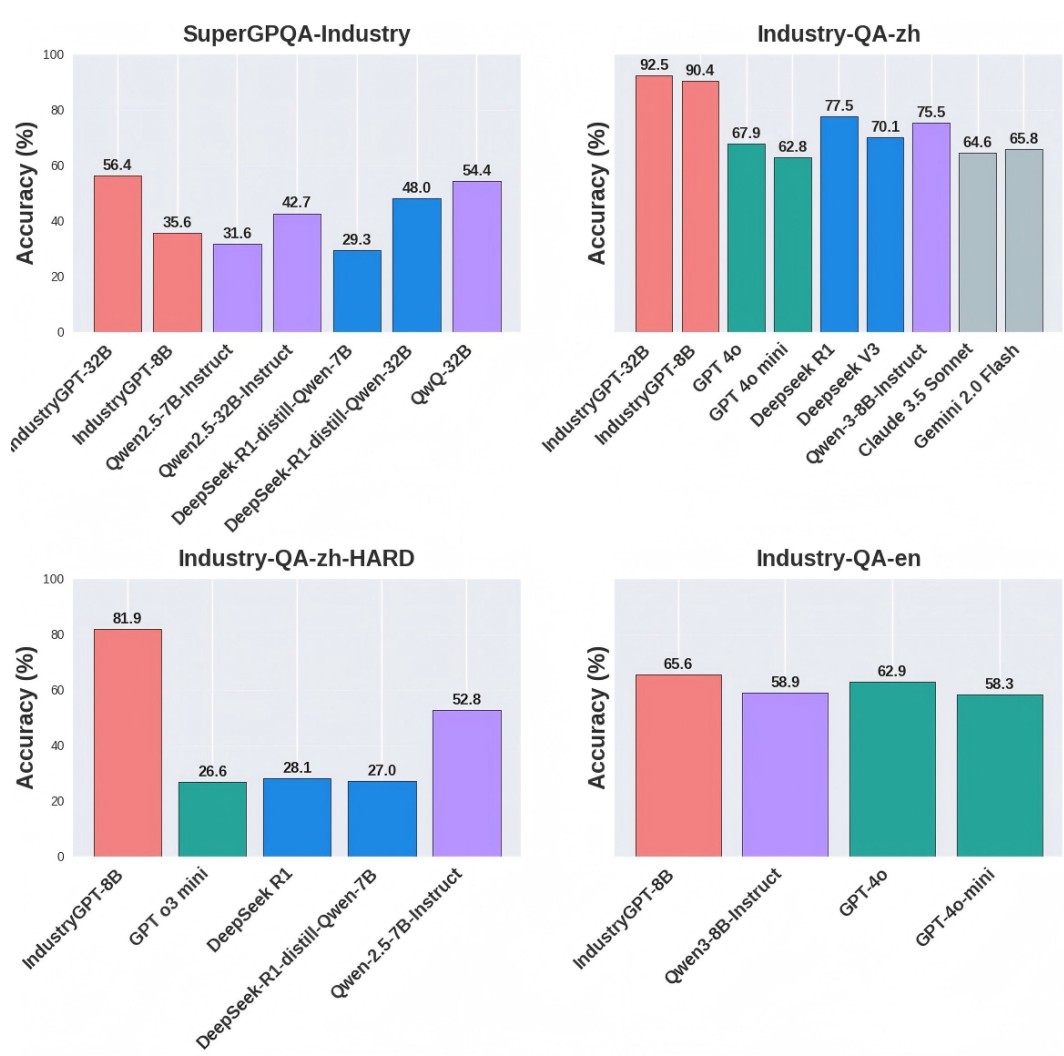

Figure 4: Overview of model performance across our four key evaluation benchmarks. IndustryGPT (muted red) consistently establishes a new state-of-the-art in the industrial domain benchmarks (SuperGPQA-Industry, Industry-QA-zh, and Industry-QA-zh-HARD) while remaining highly competitive with its base model on the general MMLU benchmark.

## 5.1 Dominant Performance on Industrial Benchmarks

Our primary results demonstrate that IndustryGPT achieves a new state-of-the-art in the industrial domain. As shown in Table 3, on our main benchmark, **IndustryGPT-32B** reaches an impressive **92.54%** accuracy in Chinese, significantly outperforming all other models.

This superior capability is even more pronounced on the **Industry-QA-Hard** subset (Table 4), which focuses on complex reasoning problems. Here, **IndustryGPT-8B** scores **81.9%**, a massive leap of over 29 points compared to its base model. This highlights the profound impact of our second-stage reasoning cultivation on solving truly difficult challenges. To confirm these findings on external data, we evaluated our models on the SuperGPQA industrial subset. The results in Table 5 show a similar trend, with our models consistently outperforming their counterparts, validating that our framework imparts robust and generalizable expertise.

Table 3: Performance on the Industry-QA Benchmark in English and Chinese (Accuracy %).

| Model Size | Model | Industry-QA (en) | Industry-QA (zh) |
|---|---|---|---|
| *8B* | | | |
| | GPT-4o-mini | 58.32 | 62.80 |
| | Qwen3-8B-Instruct | 58.90 | 80.22 |
| | **IndustryGPT-8B** | **65.61** | **90.83** |
| *32B* | | | |
| | **IndustryGPT-32B** | **68.25** | **92.54** |
| *Proprietary* | | | |
| | GPT-4o | 62.91 | 67.00 |
| | Deepseek-R1 | - | 77.50 |

Table 4: Performance on the Industry-QA-Hard (zh) subset (Accuracy %).

| Model | Accuracy (%) |
|---|---|
| Qwen2.5-7B-Instruct | 52.8 |
| **IndustryGPT-8B** | **81.9** |

Table 5: Performance on the SuperGPQA industrial subset (Accuracy %).

| Model Size | Model | SuperGPQA Acc (%) |
|---|---|---|
| *8B* | | |
| | Qwen3-8B-Instruct | 37.52 |
| | **IndustryGPT-8B** | **50.86** |
| *32B* | | |
| | Qwen3-32B-Instruct | 47.62 |
| | **IndustryGPT-32B** | **57.71** |
| *Large* | | |
| | DeepSeek-R1 | 63.24 |

## 5.2 Ablation Study: Validating the Two-Stage Framework

To dissect the contribution of each component of our methodology, we conducted an ablation study, with the results presented in Table 8. The study reveals two key insights: 1) The **Necessity of Stage 2**, as the full framework significantly outperforms the "Stage 1 Only" model, confirming that the reasoning cultivation step is crucial. 2) The **Superiority of the Decoupled Approach**, as our two-stage model also surpasses a "Single-Stage" model trained on mixed data. This empirically validates our hypothesis that a systematic, decoupled approach is more effective than a naive data-mixing strategy.

## 5.3 General Capability Preservation

A critical test for any specialized model is whether it avoids catastrophic forgetting. As shown in Table 7, the MMLU performance of our IndustryGPT models is statistically indistinguishable from their base Qwen3-Instruct counterparts. This result holds across both standard and Chain-

Table 6: Ablation study of our framework on the Industry-QA-zh benchmark (Accuracy %) using the 8B model.

| Model Configuration | Accuracy (%) |
|---|---|
| Base Model (Qwen3-8B-Instruct) | 80.2 |
| Stage 1 Only (Knowledge Infusion) | 89.0 |
| Single-Stage (Mixed Data) | 88.2 |
| **Full Two-Stage Framework** | **90.4** |

of-Thought evaluation settings, confirming that our framework successfully instills deep domain expertise without sacrificing the model's essential general reasoning abilities.

Table 7: General capability evaluation on the MMLU benchmark (5-shot accuracy %).

| Model | Standard Eval (%) | CoT Eval (%) |
|---|---|---|
| Qwen3-32B-Instruct | 70.59 | 70.64 |
| **IndustryGPT-32B** | **70.55** | **70.60** |
| Qwen3-8B-Instruct | 62.20 | 62.03 |
| **IndustryGPT-8B** | **63.11** | **63.08** |

Table 8: Ablation study of our fine-tuning framework on the Industry-QA benchmark (Accuracy %) using the 8B model.

| Model Configuration | Accuracy (%) |
|---|---|
| Base Model (Qwen3-8B-Instruct) | 80.2 |
| Stage 1 Only (Knowledge Infusion) | 89.0 |
| CoT Only | 84.9 |
| Single-Stage (Mixed Data) | 88.2 |
| **Full Two-Stage Framework (IndustryGPT-8B)** | **90.4** |

# 6 CONCLUSION AND FUTURE WORK

In this work, we confronted the nuanced challenges of adapting Large Language Models for the industrial sector. Our investigation revealed that naive fine-tuning, even with rich, explanatory data, is not a panacea and can create a detrimental trade-off between factual accuracy and reasoning generation. The two-stage SFT framework we proposed and validated offers a principled solution to this problem. By first building a solid foundation of domain knowledge before cultivating complex reasoning, our approach enables IndustryGPT to achieve a remarkable level of expertise. Our empirical results confirm this, showing that IndustryGPT sets a new state-of-the-art on industrial benchmarks, generalizing effectively to unseen academic questions while fully preserving its core reasoning abilities. This demonstrates that with the right methodology, deep specialization and broad intelligence need not be mutually exclusive.

Beyond the model itself, our contribution to the community is twofold. First, we are releasing the Industry-QA Benchmark, a large-scale and meticulously curated resource with over 10,000 questions, which we hope will foster further innovation in this domain. Second, by combining this with a standardized industrial subset of SuperGPQA, we have helped establish a more robust and comprehensive evaluation standard for what constitutes a true "industry expert" model. We believe these resources will be invaluable for future research and development, accelerating the creation of reliable and transparent AI for critical applications.

Our findings open several promising avenues for future exploration. The principles of our two-stage framework could be extended to an even wider array of industrial verticals, from aerospace to biotechnology. A particularly exciting frontier is the integration of multimodal understanding, enabling models to interpret engineering schematics, analyze diagnostic images of equipment, and truly interact with the visual language of industry. Ultimately, we believe the core methodology of decoupling knowledge and reasoning offers a powerful template for creating specialized AI in other complex, high-stakes fields. As AI continues to integrate into these critical sectors, such principled, domain-aware adaptation strategies will be paramount.

## STATEMENT ON THE USE OF LLMS IN MANUSCRIPT PREPARATION

We disclose that a large language model (LLM) was used as an assistant in the preparation of this manuscript. The LLM's role was limited to language polishing, including the reorganization of sentences for clarity and assisting in the generation of LaTeX code for the tables. However, the human authors maintained full intellectual control throughout the writing process. We hereby declare that all experimental data, the analysis of results, and the conclusions drawn in this paper are the original work of the authors, who have meticulously verified their authenticity and correctness. The LLM served strictly as a writing and formatting aid.

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

## A APPENDIX

### A.1 IMPLEMENTATION DETAILS

**Training Hyperparameters.** Our models were fine-tuned using the AdamW optimizer. Key hyperparameters for both the 7B and 32B models are detailed in Table 9. We employed a linear learning rate warmup for the first 10% of training steps, followed by a cosine decay schedule.

Table 9: Key hyperparameters used for the two-stage SFT process.

| Hyperparameter | IndustryGPT-8B | IndustryGPT-32B |
|---|---|---|
| Max Sequence Length | 2048 | 2048 |
| Learning Rate | 2e-5 | 1e-5 |
| Batch Size | 64 | 32 |
| Number of Epochs (Stage 1) | 2 | 2 |
| Number of Epochs (Stage 2) | 3 | 3 |
| Optimizer | AdamW | AdamW |
| Weight Decay | 0.01 | 0.01 |
| Warmup Ratio | 0.1 | 0.1 |

### A.2 CASE STUDY

To provide a qualitative understanding of IndustryGPT's capabilities, we present a case study on a complex industrial problem. The prompt requires both factual knowledge (properties of fluids) and multi-step diagnostic reasoning.

**Prompt:** *"A centrifugal pump in a chemical processing line is experiencing cavitation. The input pressure is 50 psi, the fluid is benzene at 80°C, and the pump is located 5 meters above the source tank. Analyze the potential causes and suggest three corrective actions."*

**Baseline Model (Llama-3-70B-Instruct) Response:**

Cavitation is caused by low pressure in the pump. This can happen if there are leaks in the suction line or if the pump is running too fast. You should check for leaks, reduce the pump speed, or increase the input pressure.

**IndustryGPT-32B Response:**

This is a classic Net Positive Suction Head (NPSH) problem. Here is a step-by-step analysis:

1. **Problem Definition:** Cavitation occurs when the local pressure within the pump drops below the liquid's vapor pressure, causing bubbles to form and collapse violently.

2. **Analysis of Benzene's Properties:** At 80°C, the vapor pressure of benzene is approximately 101.3 kPa (14.7 psi). The available pressure at the pump inlet (NPSHa) must exceed this value.

3. **NPSH Calculation:**
   - The pressure from the source tank is 50 psi.
   - There is a negative static head because the pump is 5 meters above the tank, which is approximately -7.1 psi.
   - Assuming minor friction losses (e.g., 2 psi), the total pressure at the pump inlet is roughly 50 - 7.1 - 2 = 40.9 psi.

4. **Conclusion:** The available NPSH (40.9 psi) is well above the vapor pressure of benzene (14.7 psi), so the initial parameters seem fine. *Correction:* Re-reading the prompt, 50 psi is the input pressure to the line, not necessarily the tank pressure. If 50 psi is the pressure far upstream, the static head loss is critical. The primary cause is likely the large 5-meter static lift combined with the high fluid temperature.

5. **Corrective Actions:**
   (a) **Lower the pump:** Decrease the static head by moving the pump closer to the source tank's elevation. This is the most effective solution.
   (b) **Cool the benzene:** Reduce the fluid temperature before it enters the pump to lower its vapor pressure.
   (c) **Increase source pressure:** If possible, increase the pressure in the source tank to raise the NPSHa.

The baseline model provides a generic and superficial answer. In contrast, IndustryGPT correctly identifies the core engineering concept (NPSH), performs a step-by-step analysis incorporating fluid properties, identifies the most probable cause, and even self-corrects its interpretation of the prompt, leading to precise and actionable recommendations.

