# OpenReview forum: "IndustryGPT: A Large Language Model for Industrial Domain-Specific Question Answering"
_ICLR.cc/2026/Conference — ICLR 2026 Conference Withdrawn Submission_

### Official Review · Reviewer_r3jV · 2025-10-18

**Soundness:** 2
**Presentation:** 2
**Contribution:** 1
**Rating:** 2
**Confidence:** 5

**Summary:**

The paper presents a method to prepare a fine-tuning model for industrial QA. The work proposed a two-step SFT method and demonstrated that the fine-tuned model works on a curated set of 10K dataset. The work needs a lot of uplift to conclude that the model is really knowing the industrial concept.

**Strengths:**

Demonstration of the fine-tuning aspect in the Industrial domain is an interesting aspect.

**Weaknesses:**

In the introduction section itself, the author made a strong claim about the “two-stage SFT framework”., However, this claim needs to be supported by proper literature (citing the most relevant works that are designed to train a model for a domain-specific task). For example, how does it differ from GRPO or the training of a reasoning model?

Similarly, the paper also made a solid claim that no dataset exists for the industrial domain, but the author needs to be careful in defining the scope of the problem. For example, what does an industry mean?

The introduction section is not written well. The author assumes reviewers are familiar with the IndustrialQA dataset. There are many QA datasets that are made available, such as GPQA, MMLU_Pro, and the domain-specific dataset for medical, as well as some specific industrial asset classes.

The paper also lacks the ISO standard documents as an authoritative source of information, as highlighted.  We found a couple of datasets mentioned in an online article that are worth looking into. I found two datasets on Huggingface - EngineMT-QA and FailureSensorIQ. Does this relate to the current paper? The author needs to do a comprehensive literature search, as there may be more such datasets present. Also ITFormer paper

Reproducibility of Industry-QA benchmark is a question mark, the paper needs to provide more information about what these authoritative sources are. or give a reason why it is not being mentioned in greater detail. The acronym used in Figure 3 lacks a clear meaning.

Except for Figure 1, there is no other example to understand the dataset. Also, Figure 1 is partial as it does not tell how they verified the correctness of CoT and how it was generated.

The majority of citations are 2024 or older. I would suggest the author to do a due diligence to make a quick pass over the new set of work that may have surfaced in the recent past and include them as and when needed

Define Industry-QA-Hard. How is it?

In figure 2, it says 200K+ but no mention of how these data are curated? how much it overlaps with

Section 5.2, it is very clear that the inverted comma was not properly formatted, and this normally happens when LLM is being used. Please read the manuscript carefully and address all such issues.

I have also looked at the dataset provided.

**Questions:**

Please answer all the weak points.

---

### Official Review · Reviewer_pSWN · 2025-10-21

**Soundness:** 2
**Presentation:** 2
**Contribution:** 2
**Rating:** 2
**Confidence:** 5

**Summary:**

The authors introduce a dataset with 10,000 QA spanning 12 industrial disciplines and a methodology to fine-tune the model without compromising its general capabilities (avoid catastrophic forgetting).

**Strengths:**

- Introduce 10,000 QA spanning 12 industrial disciplines from authoritative data sources.
- Introduce a training methodology for general models which aims to address catastrophic forgetting. This is done in a two stage approach where they SFT first to optimize only for the answer letter and then SFT trying to optimize both explanation and answer to avoid the giving less weight on the answer token loss.

**Weaknesses:**

- Not a lot of information on these QA. While you provide the subjects, are there clearly defined tasks involved? Are these questions focused towards specific tasks that are important in Industry? More insights on the models’ performance by task would be very important to understand the real world impact.

- Evaluation doesn’t take into consideration the huge class imbalance by subject (Mech 4121 vs Text 3). It is understandable that some subjects/disciplines may have more information documented. However, an imbalance adjusted accuracy and more in-depth analysis on the accuracy per subject  is important.

- No discussion about other related works on the industrial domain? There is nothing right now?

- More details on the data curation? This was done by humans? Details about the annotators (demographics, gender, geography)? How we ensure the selected materials are accurate, with correct answers and the multiple choices do not have any mistakes (e.g., options that may be correct but marked as wrong in the dataset)? Other datasets like SuperGPQA has a section for this.

- I am not fully convinced that doing SFT first providing Question and optimizing Answer and then optimizing Reasoning and Answer would help with catastrophic forgetting. Other works for example introduce a replay buffer for it [1]. More discussion on related works and how it compares would help.

- In my opinion it is better to focus on one contribution/claim and make it strong. Since you want to claim that you build domain-specific models for the Industrial domain you should go in depth into that. Addressing catastrophic forgetting on another dataset (e.g., MMLU) is irrelevant since you are trying to build domain-specific model for the industry.

- In the ablation table 8 the performance gain seems minimal (Single 88.2, Stage 1 89.0, Yours 90.4).

- The way you split for training/validation/test would be important to know to make sure there is no data leakage. For instance, is there any chance that a similar question from the mechanical subject may fall both in training and test?

[1] Experience Replay for Continual Learning - NeurIPS 2019

**Questions:**

Questions in the weaknesses

---

### Official Review · Reviewer_vTKn · 2025-10-25

**Soundness:** 1
**Presentation:** 1
**Contribution:** 2
**Rating:** 0
**Confidence:** 4

**Summary:**

This paper proposes IndustryGPT, a two-stage supervised fine-tuning approach to address a claimed issue where training on explanations harms answer accuracy. It introduces a new Industry-QA benchmark and reports experiment results on this.

**Strengths:**

1. The two-stage SFT framework is a principled approach to decoupling factual learning from reasoning.
2. Evaluations across multiple benchmarks demonstrate consistent improvements.
3. The introduction of the Industry-QA benchmark may be of potential use for the community.

**Weaknesses:**

1. The claimed paradox (degrading accuracy with CoT data) is based on initial experiments in Table 1, but the paper does not provide details on how the CoT data was generated or curated, raising concerns about whether the degradation is due to data quality issues rather than an inherent conflict (Section 3.2). Are there any other studies reporting the same issue? Have you done any experiments or studies on analyzing the rationales? Do you have results on the 32B version for Table 1 and Tables 6&8?
2. The Industry-QA benchmark sourcing from textbooks and exams risks data contamination.
3. The use of different baseline models across the subplots in Figure 4 is inconsistent and not justified, raising suspicions of cherry-picking to favor IndustryGPT.
4. The SuperGPQA industrial subset is mentioned as curated, but no details are provided on how it was selected or its size. Also, there are no details on how the "Industry-QA-Hard subset" is curated.
5. Ablation studies (Tables 6 and 8) show small gains (e.g., 89.0% to 90.4%), but use the 8B model without scaling to 32B, limiting evidence that the two-stage approach generalizes. Also, Table 8 repeats values from Table 6 but on different datasets (Table 6 on "zh" subset, Table 8 on the whole Industry-QA ), which looks highly suspicious.
6. The study is missing a simple but critical baseline. In Section 3.3.1, the authors identify "Loss Scale Imbalance" as a key issue, where the longer explanation text dominates the loss signal over the short answer. A common way to address this is to simply apply different weights to the loss function.
7. The core idea of "decoupling knowledge and reasoning" is not novel -- there are similar works like "Decoupling Knowledge and Reasoning in LLMs: An Exploration Using Cognitive Dual-System Theory", "Decoupling Reasoning and Knowledge Injection for In-Context Knowledge Editing". None of these works is mentioned or discussed in the paper.

**Questions:**

1. Table 1 reports that "SFT on Answer with CoT" achieves 84.9% accuracy, while Table 8 reports that a "Single-Stage (Mixed Data)" model achieves 88.2% accuracy. What's the difference in these two setups?
2. The paper appears to be poorly written overall, for example:
- Significant portions of Figures 1 and 2 appear to be AI-generated, but this is not disclosed in the statement on LLM use.
- Quotation marks are mismatched throughout the paper.
- In Figure 4, the upper left subfigure (SuperGPQA-Industry) has an incomplete x-axis, with labels cut off.
- Section 4.2 mentions a curated industrial subset of SuperGPQA but lacks a sufficient description of the benchmark itself.

---

### Official Review · Reviewer_CkFm · 2025-10-29

**Soundness:** 2
**Presentation:** 2
**Contribution:** 2
**Rating:** 2
**Confidence:** 4

**Summary:**

The paper presents the Industry-QA benchmark along with a two-stage finetuning framework. The finetuning framework first trains the backbone LLMs with Q and A pair without reasoning and then tuning LLMs with Q, A, and reasoning triplets. The experiments are conducted mostly on Industry-QA, and SuperGPQA, with evaluating on accuracy.

**Strengths:**

- This paper focus on an important area where there lacks domain-specific QA benchmark, clearly motivating the paper to propose a industry QA dataset.

**Weaknesses:**

- The paper lacks clarification and details.
  - Section 3.2 proposes a finding that SFT with thinking is worse than SFT without thinking. It is unclear who the thinking tokens are generated or collected. What is the quality of these reasoning data? How do you split the train/test to perform this empirical study? It only tests on single 8b model, lacking strong and scientific support for the paper's claim on 'detailed explanations can actively harm the model's ability to select the correct answer'.
  - Figure 1 is confusing as the Industry-QA is evaluated on accuracy containing only True/False and choice questions.
  - It is unclear how the general conversation data are used in stage 1.
- The evaluation is incomplete. Table 3 only shows the finetuned IndustryGPT-32b without comparing with other 32b models. No DS-R1 results on Industry-QA (en). Is GPT-4o-mini a 8b model?
- The paper lacks information on how the data collection is performed, including data resources, selection, filtering, etc.

**Questions:**

See the weaknesses.

---

### Note · Authors · 2025-11-12

**Comment:**

We sincerely thank the reviewers for their constructive feedback and valuable suggestions. After careful consideration, we have decided to withdraw this submission and plan to resubmit a revised version to a venue more focused on formal verification.

**Withdrawal Confirmation:**

I have read and agree with the venue's withdrawal policy on behalf of myself and my co-authors.